# The Safety of Pharmacological and Surgical Treatment of Diabetes in Patients with Diabetic Retinopathy—A Review

**DOI:** 10.3390/jcm10040705

**Published:** 2021-02-11

**Authors:** Wojciech Matuszewski, Angelika Baranowska-Jurkun, Magdalena Maria Stefanowicz-Rutkowska, Katarzyna Gontarz-Nowak, Ewa Gątarska, Elżbieta Bandurska-Stankiewicz

**Affiliations:** 1Department of Internal Medicine, Endocrinology, Diabetology and Internal Medicine Clinic, University of Warmia and Mazury in Olsztyn, 10-719 Olsztyn, Poland; angelika_b1990@o2.pl (A.B.-J.); m.m.stefanowicz@gmail.com (M.M.S.-R.); katarzyna.gontarz.92@gmail.com (K.G.-N.); bandurska.endo@gmail.com (E.B.-S.); 2Nephrology, Transplantology and Internal Medicine Clinic, Pomeranian Medicine University in Szczecin, 70-204 Szczecin, Poland; ewga@op.pl

**Keywords:** diabetes, microvascular complications, diabetic retinopathy

## Abstract

Background. Diabetes mellitus (DM) is a non-infectious pandemic of the modern world; it is estimated that in 2045 it will affect 10% of the world’s population. As the prevalence of diabetes increases, the problem of its complications, including diabetic retinopathy (DR), grows. DR is a highly specific neurovascular complication of diabetes that occurs in more than one third of DM patients and accounts for 80% of complete vision loss cases in the diabetic population. We are currently witnessing many groundbreaking studies on new pharmacological and surgical methods of treating diabetes. Aim. The aim of the study is to assess the safety of pharmacological and surgical treatment of DM in patients with DR. Material and methods. An analysis of the data on diabetes treatment methods currently available in the world literature and their impact on the occurrence and progression of DR. Results. A rapid decrease in glycaemia leads to an increased occurrence and progression of DR. Its greatest risk accompanies insulin therapy and sulfonylurea therapy. The lowest risk of DR occurs with the use of SGLT2 inhibitors; the use of DPP-4 inhibitors and GLP-1 analogues is also safe. Patients undergoing pancreatic islet transplants or bariatric surgeries require intensive monitoring of the state of the eye, both in the perioperative and postoperative period. Conclusions. It is of utmost importance to individualize therapy in diabetic patients, in order to gradually achieve treatment goals with the use of safe methods and minimize the risk of development and progression of DR.

## 1. Introduction

According to the current World Health Organization (WHO) definition, diabetes mellitus (DM) is a group of metabolic disorders characterized by the presence of hyperglycaemia caused by impaired release or function of insulin. Chronic hyperglycaemia in DM leads to damage, impairment of functioning, and insufficiency of various body organs, especially eyes, kidneys, nerves, the heart, and blood vessels [1]. At present, there are 463 million adult DM patients, which constitutes 9.3% of the world population. DM is a non-infectious pandemic of the modern world, and it is estimated that in 2045 the number of patients is going to increase to 700 million (10% of the world population) [2,3]. In line with the increasing number of diabetic patients, the number of complications accompanying DM also rises, and this study focuses on one of them, that is, diabetic retinopathy (DR). DR is a highly specific neurovascular complication of DM that appears in one third of diabetic patients [4]. DR is responsible for 80% of blindness in DM patients and can occur at any stage of the disease. [5,6]. Currently, there are many risk factors for the occurrence and progression of DR, yet there are few studies that present the impact of pharmacological and surgical treatment of DM on DR.

The aim of the study is to assess the safety of pharmacological and surgical treatment of DM in patients with DR.

## 2. Material and Methods

The study presents an analysis of data currently available in the literature that concern treatment methods of DM and their effect on the occurrence and progression of DR. The methods have been divided into pharmacological (including new antihyperglycaemic agents), surgical, and modification of lifestyle. Moreover, the study contains a brief presentation of new theories concerning the development of DR.

## 3. Results and Discussion

### 3.1. The Phenomenon of Early Worsening of Diabetic Retinopathy

The phenomenon of early worsening of diabetic retinopathy (EWDR) is a specific paradox that can occur in successful treatment of diabetes, which is characterized by deterioration of the eye condition in response to a rapid improvement in metabolic control. Both the appearance of soft exudates, haemorrhages, intraretinal microvascular abnormalities (IRMA), or microaneurysms at the fundus, as well as progression of the existing DR are characterized for EWDR [7,8,9]. The main factor in the development of EWDR consists in rapid improvement in blood glucose control, expressed in the reduction of HbA1c over a relatively short period of time. EWDR is more common in patients with poor metabolic control of diabetes at baseline and pre-existing diabetic complications of the eye. These data will be considered in detail depending on the method of diabetes treatment in the following sections [10,11]. Another potential risk factor for EWDR may consist in poor blood pressure (BP) control. The iconic UKPDS indicates that improved blood pressure control correlates with a lower risk of DR [12]. However, the Action in Diabetes and Vascular Disease-Preterax and Diamicron Modified Release Controlled Evaluation (ADVANCE), Action to Control Cardiovascular Risk in Diabetes –Blood Pressure (ACCORD BP), and more recent SUSTAIN-6 (Trial to Evaluate Cardiovascular and Other Long-term Outcomes With Semaglutide in Subjects With Type 2 Diabetes) studies showed that strict control of BP did not result in a lower risk of DR occurrence or progression. It can be said that it is a disadvantage of many studies that they employ combined assessment of the analysed risk factors [13]. It was only researchers in Singapore that assessed an individual effect of BP on DR. The study included 2189 DM patients with concomitant arterial hypertension. In the whole analysed group, the prevalence of any-DR was 33.8% and the vision-threatening DR (VTDR) 9.0%, the mean systolic blood pressure (SBP) was 157.4 (19.4) mmHg, and diastolic blood pressure (DBP) was 83.6 (11.2) mmHg. Both poorly controlled and untreated hypertension were significantly associated with any-DR. Moreover, SBP and pulse pressure were associated with any-DR and VTDR [14]. In the literature of the subject, attention is also paid to disorders of lipid metabolism in the context of DR development. It has been shown that in patients treated with insulin, higher cholesterol levels were related to a more frequent occurrence of hard exudates, and each 10% increase in cholesterol levels increased the number of hard exudates by 50% [15,16]. The number of hard exudates was reduced as a result of implementing statin treatment [17]. The Diabetes Control and Complications Trial (DCCT) revealed a positive correlation between the advancement of DR and increased levels of triglycerides (TG) and decreased levels of high density lipoprotein cholesterol (HDL) [18]. Additionally, Zhang proved that increased VLDL (very low density lipoprotein) and TG levels are independent factors contributing to an increased DR risk [19]. Australian researchers have shown that apolipoproteins may be more accurate markers of DR compared to the traditional lipid profile. A study including 224 DM patients demonstrated that only lower HDL cholesterol was independently associated with diabetic retinopathy. However, serum apolipoprotein: apoAI, apoB, and the apoB-to-apoAI ratio were consistently associated with the presence of DR and severity of DR, independently of other DR risk factors [20]. The role of lipoprotein (a), whose higher circulating levels correlated positively with DR, is also discussed [21]. However, no such relationship was found in other studies [22].

Below, we present an original acronym that corresponds to modifiable risk factors for the development of EWDR, which should be taken into account in everyday clinical practice (Figure 1).

Prevention of EWDR comprises patient education and step-by-step implementation of metabolic goals in diabetes treatment, which are related to glycemia, lipids, or blood pressure. When diabetes treatment is intensified, more frequent visual assessment is recommended—every 3–6 months, depending on the baseline fundus condition. Feldman-Billard et al. proposed a practical management algorithm in the period of intensified diabetes treatment ([23], Figure 2).

### 3.2. Mechanisms of EWDR Development

Before analyzing particular pharmacological and surgical methods of treating DM, it can be asked in which pathophysiological mechanisms DR occurs and develops. There are a number of theories concerning this issue; however, the EWDR pathomechanism has not been fully comprehended as of yet.

The osmotic force theory assumes that glucose is an osmotically active molecule. A change in the glucose concentration alters the osmotic pressure and water retention. Hyperglycaemia lowers the intra-vascular osmotic pressure, and the change in the osmotic gradient inside the cell compared to that inside the vessel results in fluid displacement, excessive stretching of the vessels, and the formation of microaneurysms. It is a disadvantage of this theory that it does not rationally account for a higher frequency of DR in patients treated with insulin compared to those on oral antidiabetic agents (OADAs) and the fact that it does not explain increased angiogenesis and neovascularization [24,25,26,27,28].

Another theory concerns the effect of insulin on retinal haemorrhages. An in vivo study scrutinized the effect of insulin on the development of oxygen-induced retinopathy in seven-day-old rats in a 55% oxygen environment for day days. Normal saline or human insulin was administered subcutaneously with or without riluzole (an anti-amyotrophic lateral sclerosis drug) at normal oxygen levels. The insulin group was found to exhibit higher levels of vascular endothelial growth factor (VEGF), vascular carcinogenesis, greater activity of matrix metalloproteinases, and a greater area of retinal haemorrhages. Riluzole significantly reduced all the insulin-induced changes [29].

In contrast, the synergistic theory is based on the fact that insulin is an anabolic hormone. Under conditions of rapid glycaemic reduction, high doses of exogenous insulin may affect retinal hypoxia synergistically with VEGF, causing vascular proliferation and progression of retinopathy [30]. Insulin interacts with Nox-4 (NADPH oxidase subunit 4), increases the production of ROS (reactive oxygen species), and in turn increases the oxidative stress in the retinal endothelial cells. ROS produced by insulin activates HIF-1α (hypoxia-inducible factor 1α), increasing VEGF expression, which results in activation of angiogenesis, neovascularization, and damage to the retinal–blood barrier [31,32]. Participation of VEGF in DR development is well documented, and VEGF inhibitors currently constitute the basic PDR therapy [33].

The role of growth hormone (GH), insulin-like growth factor (IGF-1), and insulin-like growth factor-binding protein (IGFBP) is also discussed in DR pathogenesis. Both insulin and IGF-1 have an impact on retinal neovascularization by modulating the expression of various vascular mediators [34,35]. Studies on mice in an anoxic environment have shown that IGF-1 over-expression leads to the development of proliferative DR [36]. Similarly, after the analysis of patients with Laron syndrome, which consists in genetic defects in the GH/IGF-I axis, pathological changes in retinal vascularization dependent on low IGF-1 were demonstrated. Compared to the control group of 100 healthy volunteers, a statistically significantly lower number of vascular branching points was demonstrated in patients with the GH/IGF-1 axis defect (median 23, range 16–25 vs. median 28, range 19–40, *p* < 0.001) [37]. However, in a study conducted in 2001–2009 on 480 DM2 patients, Chen did not find a statistically significant correlation between an elevated IGF-1 level and DR progression. It has been concluded that IGF-1 may be a factor promoting progression from NPDR to PDR masked by poor metabolic control of diabetes [38]. The GH/IGF-1 axis theory did not, however, gain support after the publication of results of a study on 25 DM1 and DM2 patients who had received subcutaneous injections of pegvisomant for 12 weeks. Although the serum IGF-1 level decreased by 55% on average, no regression of diabetic retinal neovascularization in any patient was observed. The fundus status did not change in 16 patients, and progression of lesions was observed in nine remaining patients [39].

Other experimental and clinical observations reported metabolic memory in DM patients, which is based on cells, mainly the endothelium, early exposure to a hyperglycaemic environment, and which may be the cause of vascular complications in DM, when glycaemic control has not been achieved very early [40,41,42]. To explain the phenomenon of metabolic memory, it may be necessary to refer to epigenetic mechanisms, which link genetic and environmental factors. In DR, the histone hypomethylation of H3K4 occurs with a down-regulation of gene expression for the antioxidant enzyme superoxide dismutase 2 (SOD2) [43,44,45]. Hyperglycemia reduced levels of H3K4me1 and H3K4me2 and increased the binding of lysine-specific demethylase-1 (LSD1) and Sp1 at the SOD2 gene promoter. The histone codes H3K9ac, H3K12ac, H3K4me2, and H3K4me3 suppress transcription of endothelial nitric oxide synthase (eNOS) and impair NO production [46]. In DR, an increased activity of matrix metalloproteinases (MMPs) was found, which was also shown in the retina of diabetic mice [47,48]. As already shown, MPP-9 activation in the retina causes the development of DR [49,50]. The possibility of exploiting the reversibility of epigenetic changes and targeting enzymes important for histone methylation may serve as a potential therapy to halt the development of DR and normalize glucose metabolism.

Epigenetic drugs are small molecules with epigenetic and anti-diabetic activity. Intense research is undertaken to investigate the enzymes responsible for epigenetic modifications. The enzymes modify chromatin: histone acetyltransferase inhibitors (HATi), histone deacetylase inhibitors (HDACi), HDAC activators, and miRNA inhibitors. Interference with RNA-based mechanisms is also used [51]. Among effective HAT inhibitors, there is garcinol, from garcinia fruit rinds, whose targets are P300/CBP-associated factor (PCAF) and p300. Garcinol was able to reduce inflammatory proteins in retinal Müller glia, which increased at a high-glucose concentration [52,53]. Curcumin, the active compound of Curcuma longa (turmeric), exhibits DNA hypomethylating activity and influences both histone acetylation and miRNA expression. Curcumin could ameliorate glucose metabolism and could also prevent diabetic complications by increasing HDAC-2 and decreasing p300-HAT in human monocytes with a consequent reduction of Nuclear Factor kappa-light-chain-enhancer of activated B cells (NF-kB) signaling and vascular inflammation [54]. It was shown that curcumin increases postprandial serum insulin levels without affecting plasma glucose levels in healthy subjects [55]. Additionally, it was proved that the multicomponent nutritional formula ameliorates glycemic control, visual function, and peripheral neuropathy in DM patients [56]. Evaluation of epigenotypes by epigenome-wide association studies (EWAS) is a method that provides new information on the pathogenesis of diabetic complications and metabolic memory. Epigenetic traits may represent new targets for individualized therapy to increase the survival with a lower risk of toxicity [57].

Novel therapies based on epigenetic modulators, emerging from innovative technologies, might help reduce the global burden of DM and it complications. Mechanisms involved in reversible epigenetic changes increased the possibility of developing new therapeutic strategies and might represent an additional opportunity in combination with standard anti-diabetic treatments. Summarizing the above chapter, we present potential mechanisms of EWDR development (Figure 3).

### 3.3. Pharmacological Treatment

#### 3.3.1. Insulin

In 2021, the 100th anniversary of the discovery of insulin will be celebrated. On 23 January 1922, insulin saved a patient’s life for the first time, the lucky one being Leonard T. Since then, insulin has been modified many times through genetic engineering to improve the length and quality of life [58]. However, in the 1980s, the first reports revealed deterioration of the eye condition in patients with diabetes treated with personal insulin pumps. This phenomenon was referred to as the early worsening phenomenon in DR. In this context, early refers to the achievement of rapid glycaemic control, not to the short duration of diabetes. One of the first publications on this subject was an observation from 1892 concerning 18 patients treated with personal insulin pumps. Within 3–6 months of intensified treatment, seven of them experienced deterioration of DR, and four progression from moderate to severe non-proliferative diabetic retinopathy (NPDR) [59]. In the following years, further reports were published in the Krock Collaborative Study Group; after eight months of follow-up, deterioration of DR was confirmed in 47% of patients treated with personal insulin pumps, compared to 27% in the conventional treatment group [60]. Similarly, in the Oslo Study, after three months of functional intensive care (FIT) conducted with the use of insulin pumps and the method of multiple subcutaneous injections of insulin, 50% of patients exhibited deterioration of DR, which was not observed in the conventional treatment group. However, after two more years of follow-up, more microaneurysms and hemorrhages were recorded in the conventional group rather than in the FIT group [8]. In the renowned DCCT (Diabetes Control and Complications Trial), 1441 DM1 patients were analyzed; after 6–12 months, the frequency of DR occurrence was higher in the intensive care group compared to the conventional treatment group, and amounted to 13.1% vs. 7.6% (*p* < 0.001). The risk factors for DR worsening in this trial were the presence of previous eye fundus lesions (soft exudates, intraretinal microvascular abnormalities (IRMA)), a high HbA1c level at the screening visit, and high HbA1c reduction within 0–6 months. However, in a 10-year follow-up, the risk of DR progression was significantly lower in the FIT group [61]. The presented studies illustrate how the model of administered insulin therapy affects the risk of DR. However, it was a meta-analysis of seven studies from 1996–2014 that showed how insulin itself affects this risk. The studies involved 19,107 patients with DM2, including 1711 patients with already diagnosed DR. Based on these studies, it was found that insulin therapy statistically significantly increases the risk of DR (OR 2.30; 95%; Cl 1.35–3.93) [62]. In a prospective observational study of DM2 patients treated with insulin, a positive correlation was found between the IGF-1 (insulin-like growth factor 1) level and DR progression [63]. Insulin is an anabolic hormone, and the multiple role of IGF-1 in retinal neovascularization and, consequently, the development of DR has been proven in a number of studies [64,65,66,67]. This raises the question of safety when inulin preparations currently available on the market are used in DR. In a comparison of NPH (neutral protamine Hagedorn) insulin administered twice a day to glargine administered once a day, no difference was found in the risk for the occurrence and progression of DR. However, treatment with NPH insulin was associated with a higher frequency of severe hypoglycaemia compared to insulin glargine [68]. Insulin therapy leading to a rapid reduction of glycemia soon brings the risk of deterioration of the eye condition. Thus, intensification of insulin therapy and achievement of glycaemic goals should be performed gradually. Thanks to their safe action profile, the use of new ultra-long insulin analogues (glarginU300, degludec) seems to carry a low risk of DR development, but their application requires further studies.

#### 3.3.2. Metformin

In accordance with the current guidelines of diabetes societies, metformin is still the drug of first choice in the treatment of DM2 [69,70]. Numerous reports, such as the United Kingdom Prospective Diabetes Study (UKPDS), show that the use of metformin statistically significantly reduces the risk of many diabetes complications, including retinopathy [71]. Current research also highlights the safety of administering metformin. Yue Li et al. demonstrated a statistically significant reduction in the risk of DR progression in patients with long-lasting DM2 (≥15 years) who had been on metformin for at least five years [72]. Christina Ryu et al. showed that proliferative diabetic retinopathy (PDR) developed to a lesser extent in patients with long-lasting DM 2 (≥20 years) who had been on metformin compared to a group without metformin, 27.3% vs. 45.5%, respectively [73]. The positive effect of metformin on the retina is explained by the inhibition of angiogenesis and reduction of retinal inflammation [74]. Moreover, in mice, metformin was shown to reduce VEGF 2 receptor phosphorylation, which inhibits VEGF signaling responsible for neovascularization of the retina [75]. Another study determined that metformin can inhibit the translation of the VEGF-A protein by inducing the VEGF-A-targeting microRNA, microRNA-497a-5p, which also results in reduction of retinal neovascularization [76]. Thus metformin is not only a safe drug, but its use is clearly associated with reducing the risk of DR occurrence and progression.

#### 3.3.3. Sulfonylurea Derivatives

Sulfonylurea (SU) derivatives were discovered in 1942 by Janbon and his team, who observed their hypoglycaemic effect in animal models [77], yet they were used to treat DM2 as late as the 1960s. The first generation SU derivatives (tolbutamide, chlorpropamide) are no longer applied, only their second generation is currently in use: gliclazide, glipizide, glibenclamide, and glimepiride. Thanks to the low cost of therapy, despite the known side effects and constantly limited indications, they are still often administered. SU derivatives act through SUR1 and its SUR2 isoform on the surface of beta cells of the pancreas and other tissues (the heart, the retina, bones, muscles), regulating the flow of K ions, and secondarily Ca ions, through ATP-dependent channels. The main effect of SU derivatives is an increase in insulin levels, which also entails side effects such as hypoglycaemias and weight gain. Additionally, SU derivatives perform a number of extra-pancreatic activities—they influence lipid metabolism, water balance, inhibition of platelet aggregation, and myocardial contractility. SU derivatives constitute a heterogeneous group of substances showing no class effect, and its individual representatives should be considered separately [78,79]. In a Japanese study from 1983, in 289 patients with DM2, gliclazide was shown to be safer than glibenclamide, which was responsible for more frequent changes in the eye fundus [80]. Another 1983 Japanese study showed a lower frequency of DR in the group treated with gliclazide (8%) compared to glibenclamide (31%) [81]. However, the Diabetic Retinopathy Program (DRP), which included a larger study group and a longer follow-up period (24 months), did not show such significant differences between individual SU derivatives in the assessment of DR risk [82]. Data concerning the safety related to the use of SU derivatives mostly go back to the 1980s. However, in 2018, a retrospective evaluation of 36 independent clinical trials was published, in which 100,928 DM2 patients participated in total, including 1806 patients with already diagnosed DR. The study group consisted of patients with an average diabetes duration of 8.7 years, characterized by poor metabolic control of HbA1c 8.2% (SD 0.5%). The analysis did not reveal any significant correlations between retinopathy and new groups of antihyperglycaemic drugs. Only the use of SU derivatives indicated a statistically significantly higher risk of DR compared to placebo (OR 1.67; 95% CI 1.01–2.76) [83]. Hyperinsulinemia and hypoglycaemia may prove to be the mechanisms responsible for the occurrence of DR. Gliclazide seems to be the safest molecule in this group, as it acts only through SUR1, and its use is associated with the lowest risk of hypoglycaemia [77]. Moreover, it inhibits platelet aggregation and adhesion. Additionally, by increasing the activity of tissue plasminogen activator (tPA), it increases the fibrinolytic activity of the vascular endothelium [84,85].

#### 3.3.4. Thiazolidinediones

Thiazolidinediones (TZDs) are agonists of peroxisome proliferator-activated receptor (PPARγ) found in adipose tissue as well as macrophage cells, endothelium, beta-cells of the pancreas, liver, heart, lungs, and skeletal muscles. TZDs were introduced into the treatment of diabetes in 1997, and thanks to their strong effect on reducing insulin resistance, they were initially believed to be drugs that could cure DM2 patients. However, currently they are used only in a few groups of patients due to their serious side effects [86,87]. Troglitazone was the first representative of this group of agents, yet it was withdrawn from the European market after a few weeks as a consequence of its hepatotoxicity and an increased risk of liver cancer. Rosiglitazone and pioglitazone were introduced in Europe in 2000, and a year earlier in the USA [88]. In a retrospective nine-year study involving 103,368 DM2 patients without diabetic macular oedema (DMO), the use of these two drugs was associated with an increased risk of DMO occurrence [89]. Similarly, in 2006, after an analysis of 996 new DMO cases, it was found that the use of glitazones may contribute to the development of this complication [90]. In another study published in 2008, Shen et al. analyzed 292 DM2 patients and did not confirm any effect of rosiglitazone on DMO [91]. Subsequently to various reports on the safety of TZDs in relation to diabetic eye disease, a comprehensive post hoc analysis of the ACCORD (Action to Control Cardiovascular Risk in Diabetes) study was published in 2010. In the ACCORD-Eye study, 3537 DM2 patients were qualified, among whom 695 (20.0%) were treated with TZDs and 6.2% were diagnosed with DMO. The use of TZD improved visual acuity (*p* = 0.0091) and was not associated with an increased risk of DMO [92]. Findings concerning ocular effects of TZDs are contradictory, and increased fluid retention may be believed to be a mechanism responsible for DMO development.

### 3.4. New Antihyperglycaemic Drugs

#### 3.4.1. Incretin Agents

##### I-DPP4

In 1964, in London and Denver, the glucose-dependent incretin effect was demonstrated, which gave rise to new research and contributed to the discovery of new drugs to administer in the treatment of DM2 patients [93]. Sitagliptin appeared on the market in 2006, followed by more molecules. Dipeptidyl peptidase-4 DPP-4 inhibitors have undergone a series of trials (vs. placebo) assessing their safety: saxagliptin in the Savor-timi, alogliptin in Examine, sitagliptin in Tecos, linagliptin in Carolina; vildagliptin also obtained positive cardiac safety results [94,95,96,97,98]. In a study performed on rats, Goncalves demonstrated a protective effect of sitagliptin on the vascular endothelium and the blood–retinal barrier [99]. Another study showed that sitagliptin increases the expression of the VEGF gene [100]. Ott determined that saxagliptin reduces blood flow through fine vessels of the retina and improves vasodilatation [101]. Analyzing the database of the National Health Insurance Service from South Korea, a group of 14,552 DM2 patients treated with DPP-4 inhibitors was identified, and in the first year of treatment a statistically significantly higher risk of ocular events (occurrence of vitreous haemorrhage, the need for vitrectomy, photocoagulation or intravitreal injections, or the occurrence of blindness) was confirmed compared to patients with DM2 not using iDPP-4. However, no such relationship was observed in patients using iDPP-4 for more than 12 months [102]. Another Korean study showed that the lowest risk of developing DR occurs after adding a DPP4 inhibitor to metformin, compared to SU derivatives or TZDs [103]. In an in vitro study, linagliptin had a protective effect on the retinal vascular endothelium by suppressing TNF-alpha and anti-inflammatory activity [104]. The above studies support the claim about the safety of using DPP4 inhibitors in patients with DR.

##### GLP-1RA

A glucagon-like peptide 1 (GLP-1) molecule was discovered in 1983, but the first drug in this group—exenatide—appeared on the American market only in April 2005 [105,106]. A breakthrough in GLP-1 research came with the LEADER study (Liraglutide Effect and Action in Diabetes: Evaluation of cardiovascular outcome Results), which was enthusiastically received because of the excellent results in reducing cardiovascular events exhibited by liraglutide. The same study found that DR incidents were more frequent in the liraglutide group compared to the placebo group, but statistical significance was not achieved (0.6 vs. 0.5 events per 100 patient-years; hazard ratio (HR), 1.15, 95% CI, 0.87 to 1.52; *p* = 0.33) [107]. Another representative of GLP-1 analogues—semaglutide—in the SUSTAIN 6 study concerning 3297 DM2 patients showed a statistically significantly higher risk of DR incidents compared to the placebo group, 3.0% vs. 1.8% (1.5 vs. 0.9 events per 100 patient-years; HR, 1.76, 95% CI, 1.11 to 2.78; *p* = 0.02) [108]. The risk of developing ocular complications in the semaglutide group was higher in patients with long-term diabetes, higher baseline HbA1c, a prior diagnosis of DR, and those treated with insulin. The occurrence of ocular complications in the SUSTAIN 6 study was probably caused by a rapid and significant reduction in blood glucose levels during the first 16 weeks of the study (a reduction in HbA1c was as high as 2.5% in the 1 mg semaglutide group for patients with events, with −1.4% as the mean score for the whole analyzed population over the course of 104 weeks of the entire study). Earlier studies, SUSTAIN 1–5 and a Japanese study of 3150 patients treated with semaglutide, showed no increased risk of ocular adverse events [109]. At the 56th European Association for the Study of Diabetes (EASD) congress on 21–25 September 2020, a study describing the use of semaglutide eye drops was presented. In a mice model, droplet semaglutide was shown to counteract neurodegeneration, neuroinflammation, and vascular leakage within the retina [110]. In other cardiovascular safety studies concerning GLP-1 agonists (with long follow-up of a large population of DM2 patients), neutral effects regarding ocular complications were observed. In the Exenatide Study of Cardiovascular Event Lowering (EXSCEL) trial with long-acting exenatide, ocular events were not statistically analyzed, they were reported after randomization, and in the group with the analyzed drug (7356 people), blindness occurred in eight patients, while other complications concerned 201 patients; in the placebo group it was, respectively, nine and 195 people during five years of follow-up [105]. In the Harmony Outcomes study with albiglutide, ocular events characterized as new diabetes-related blindness, laser photocoagulation, or anti-vascular endothelial growth factor treatment or vitrectomy for diabetic retinopathy were found in the number of 41 in the albiglutide group (4731 people) and 60 in the placebo group during 1.6 years of follow-up [111]. In turn, in the Researching Cardiovascular Events With a Weekly Incretin in Diabetes(REWIND) study with dulaglutide, ocular events such as secondary endpoints were reported in 95 in the dulaglutide group (4949 patients) and 76 in the placebo group during 5.4 years of follow-up [112]. The above data for GLP-1 agonists are based on standardized reporting of adverse event, so it is necessary to wait for results of the FOCUS study with semaglutide (a drug with the highest antihyperglycaemic potential), in which evaluation of its effect on ocular complications will be based on retinal imaging [113,114]. In 2017, a meta-analysis of 37 studies with GLP-1 receptor agonists was published, scrutinizing 21,782 patients treated with GLP-1RAs compared to 17,296 patients treated with other antihyperglycaemic drugs. It turned out that the use of GLP-1RAs did not have a statistically significant effect on the increase in the frequency or progression of DR (MH-OR [95% CI] 0.92 (0.74 to 1.16); *p* = 0.49), and their use was associated with a lower risk of DR than therapy with SU derivatives [115]. Similarly, in the elderly population, no relationship was found between the use of incretin drugs: both DPP4 and GLP 1 were associated with an increased risk of DR [116]. In 2019, results of the American FDA (Food and Drug Administration) report on adverse events (Adverse Event Reporting System—FAERS database) were published during the use of GLP1RAs in relation to the occurrence of DR, MDO, PDR, vitreous haemorrhage, or blindness. The report for the period between 28 April 2005, and 30 October 2017, concerned 389 ocular events, including 263 related to exenatide, 82 liraglutide, 16 albiglutide, and 28 dulaglutide. Results of the report support the claim about the safety of GLP1 receptor agonists; no increased risk of developing DR was observed [117].

#### 3.4.2. Flozins

Flozin was isolated from the bark of apple tree roots as early as 1835, but it was not until 2012 that dapagliflozin had become the first representative of this group of drugs to be approved for the treatment of DM2 in the EU [118]. Recommended by diabetologists as a second-line therapy, in 2019 it was, however, determined by the European Society of Cardiology as the gold standard in addition to GLP-1 in patients with cardiovascular risk [119]. Sodium-dependent glucose transporters (SGLTs) and facilitative glucose transporters (GLUTs) regulate the cellular uptake of glucose molecules, also in the retina [120]. SGLT-2 inhibitors reduce elevated sodium-dependent glucose uptake, which causes intracellular oedema of pericytes leading to loss of their function and their death [121]. By removing excess glucose from the retinal microcirculation, they reduce glucotoxicity, oxidative stress, and inflammation [122]. Moreover, an interaction between SGLT2 and the sympathetic nervous system (SNS) was proven [123]. SGLT2 inhibitors counteract over-activation of the SNS and prevent damage to the nerve fibers of the outer retinal layers [124]. A post hoc analysis of the Empagliflozin Cardiovascular Outcome Event Trial in Type 2 Diabetes Mellitus PatientseRemoving Excess Glucose(EMPA-REG OUTCOME) study assessed the combined endpoint for DR treatment encompassing the need for retinal photocoagulation, intravitreal drug administration, the occurrence of hemorrhage, or blindness. The intake of empagliflozin was not associated with an increased risk of the assessed DR events compared to the placebo group (1.6% vs. 2.1%, *p* = 0.1732) [125]. On the other hand, Ott et al. demonstrated a positive effect of dapagliflozin on retinal vascular remodeling, expressed as an increased wall-to-lumen ratio in the placebo group [126]. Following a meta-analysis published in 2018, covering over 100,000 DM2 patients, flozins are now considered, along with GLP-1RA and DPP-4, to be the safest form of therapy in terms of the risk of DR development [83]. The i-SGLT2 registration studies as well as the empagliflozin outcome trial in patients with chronic heart failure with reduced ejection fraction (EMPERROR) and dapagliflozin in patients with heart failure (DAPA-HF) studies have shown a spectacular reduction of vascular complications affecting the heart and kidneys. Hopefully, in subsequent studies, flozins will achieve similar results when it comes to the eye [127,128]. Summarizing, below we present comparisons of risk of diabetic retinopathy events associated with glucose lowering drugs in patients with DM (Figure 4).

### 3.5. Surgical Treatment of Diabetes

#### 3.5.1. Pancreatic Islet Transplantation

Among methods of treating diabetes, apart from the pharmacological approach, pancreas transplantation and pancreatic islet transplantation (PIT) are becoming more and more common. Since the first PIT in 1974 performed by Najarian and Sutherland, the number of such transplants has been steadily increasing. Data on such procedures are collected in the CITR (Collaborative Islet Transplant Registry), which includes data from North America and partially from centers in Europe and Australia. The 10th CITR Report of 2017 shows the number of transplant recipients from 1999–2015, which is 1086 people. Ocular side effects such as eye disorder, ocular surface disease, retinal detachment, retinal haemorrhage, uveitis, and vitreous haemorrhage were reported in 21 patients out of 877 after an ITA (Inslet Transplant Alone) surgery, in two out of 24 after SIK (Simultaneous Islet Kidney), and in six out of 183 after IAK (Islet After Kidney). The registry does not contain data on the condition of transplant recipients’ retinas before transplantation [129]. In 2005, Lee et al. described a group of 12 DM1 patients who underwent PIT. There was no progression of DR in the entire analyzed group, and one patient improved one year after the transplantation [130]. Researchers from Seoul analysed 153 diabetic patients (88 women) (79% DM1, 21% DM2), with the mean age of 36.2 ± 10.7 years, for the risk of diabetic eye disease. Initially, before PIT, DR was found in 72.9% of the entire study group. Within 12 months after PIT, 18.8% of patients experienced progression of DR, and 4.2 years after PIT, it was 20.5% of patients. Thus, the researchers themselves recommend careful monitoring of the ocular fundus condition in patients in the perioperative period and in the years following PIT [131]. Much more disturbing results came from a Chinese study that assessed six women with DM1 with an average duration of 9.8 years. The glycaemic control in these women was poor prior to PIT, their mean HbA1c was 13.4%, and mild NPDR was found on the fundus examination. Two months after the PIT procedure, HbA1c was re-measured, being 6.5% on average, and all patients had acute, symptomatic form of DMO and peripherally soft exudates. In three of them, treatment with bevacizumab injections was necessary, and in the remaining three, after six months, spontaneous healing took place [132]. That is why patients after PIT should undergo periodic ophthalmological examinations. In 2020, reports were published on prospective observations stating that EWDR occurs in a significant percentage of DM1 patients treated with SPKT (simultaneous pancreas–kidney transplantation). Progression of DR grade was assessed based on the occurrence of a composite endpoint (new need for laser therapy, newly diagnosed proliferation, macular edema, visual acuity worsening, and/or blindness over 12 months). Ophthalmological evaluation was performed before the transplantation and six and 12 months after surgery. In a group of 43 recipients age 27–65 with both functional transplants, all of whom achieved normoglycaemia in the first week after transplantation, no case of sudden progression of DR was observed, and within one year, composite endpoint occurred in 37% of recipients. According to the authors of the study, the deterioration of the DR grade seems to be related primarily to the evolution of changes occurring before the surgery, not due to rapid normalization of glucose metabolism. Researchers do not dispute the importance of metabolic control in pancreatic transplant candidates, but they do not associate ocular changes after transplantation with differences in diabetes control before and after surgery. Normoglycaemia achieved as a result of surgery has a positive impact on the further course of previously diagnosed DR, which may result in its stabilization or improvement [133].

#### 3.5.2. Metabolic Surgery

Currently, we witness great progress in bariatric treatment, i.e., obesity surgery, also referred to as metabolic surgery. The most common procedures are: gastric banding (GB), sleeve gastrectomy (SG), and a surgery to bypass (reduce) the stomach according to Mason—Roux-en-Y Gastric Bypass (RYGB) [134,135]. In diabetic patients, instantaneous correction of blood glucose levels, along with post-operative postprandial hypoglycaemia (more common after RYGB and SG procedures), potentially contributes to worsening DR. Better understanding of the complex pathophysiology of postprandial hypoglycemia after bariatric surgery and introducing an appropriate therapy may prove to be a key to improving the condition of the retina [136]. In the study, Chen et al. analyzed whether bariatric surgery might influence DR progression. The study involved 102 patients with DM2; before surgery, 1% of them had PDR, 31% had NPDR, and the fundus was normal in the remaining ones. However, after surgery, 19% of the study participants developed a new DR, 70% experienced a stable course of DR, and in 11% DR improved. It was concluded that the risk of DR progression is the highest in young men with preoperative DR and poor metabolic control, and bariatric surgery does not prevent DR [137]. A similar issue was also analyzed by Murphy et al. [138] on the example of 318 DM2 patients after bariatric surgeries. Before the surgery, 4% had PDR, 27.4% had NPDR, an in the remaining cases the fundus was normal. After the surgery, the condition of the eye was reassessed and DR progression was observed in 16%, DR improvement in 11%, and the course of the remaining DR cases was stable. Risk factors for DR progression turned out to be a higher degree of DR advancement before and a large reduction of HbA1c after the procedure. More detailed and more frequent monitoring of DR after bariatric surgery was also postulated [138]. Summarizing the mutual correlation between DR and metabolic surgery, it is worth quoting results of a meta-analysis of 10 studies from 2013–2016. Bariatric surgery was performed on 2966 patients with DM2, whose mean age was 48.4 ± 3.88 years, while mean diabetes duration was 6.7 ± 3.02 years; all participants were obese (BMI 44.8 ± 4.81 kg/m^2^), with metabolically decompensated diabetes (HbA1c 8.4 ± 3.2%) and mean values of systolic BP 140.2 ± 7.91mmHg, and diastolic BP 85.7 ± 5.57 mmHg. Following the postoperative analysis of patients, the BMI (34.4 ± 5.08), HbA1c (7.0 ± 3.1%), and blood pressure (BPs 131.6 ± 1.39 mmHg, BPd 79.3 ± 1.81 mmHg) all improved, but there was no statistically significant correlation between these parameters and the frequency or progression of DR [139]. Patients undergoing bariatric surgeries require intensive monitoring of the eye condition, both in the perioperative and postoperative periods.

### 3.6. Lifestyle Changes

A fundamental element of treating patients with diabetes is them changing their current lifestyle, which is understood as the use of nutritional treatment, regular physical activity, and stimulant use cessation. A post hoc analysis of the DCCT showed that a high-calorie diet, especially rich in fatty acids, as well as smoking are statistically significantly correlated with a higher risk of DR progression [140]. Interesting conclusions were also drawn based on an analysis of 522 pre-diabetic patients with an average age of 55 conducted five years after the conclusion of the Finnish Diabetes Prevention Study. It appeared that lifestyle changes resulted in a significantly lower incidence of retinal microaneurysms, and hypertriglyceridemia turned out to be an independent factor in the development of early microangiopathic changes [141]. Another study involved 240 DM2 patients with ocular complications (80 patients with mild-to-moderate NPDR, 80 with severe-to-very-severe NPDR, and 80 with PDR), while the control group consisted of 80 healthy volunteers. A higher level of physical activity correlated with a lower level of DR, regardless of the HBA1c level and BMI [142]. Similar conclusions were presented in a meta-analysis of 22 studies published in 2019. Regular physical activity was associated with a statistically significantly lower risk of occurrence and progression of DR [143]. Lifestyle change is an essential part of treating all patients with diabetes. In addition to proper nutrition and physical activity, what is also important is stimulant use cessation, optimal sleep time and quality, avoiding stressful situations, and regular education. Without these elements and the patient’s engagement, even if modern therapy is implemented, the goals of diabetes treatment will not be achieved.

## 4. Conclusions

We live in a period of the COVID-19 pandemic caused by the SARS-CoV-2 virus, but in these difficult times it is crucial not to forget about diabetes, which has been recognized by the WHO as the first non-infectious pandemic of the modern world. Modern individualized treatment of a patient with diabetes requires not only achieving glycaemic goals but, above all, reducing mortality and improving the quality of patients’ lives. These goals can be achieved by counteracting the occurrence and progression of macro- and microvascular complications of diabetes, including DR. The safety of the administered therapy should be doctors’ priority, and information about its benefits and risks should be included in each of the modern diabetes treatment algorithms proposed by global diabetes societies.

## Figures and Tables

**Figure 1 jcm-10-00705-f001:**
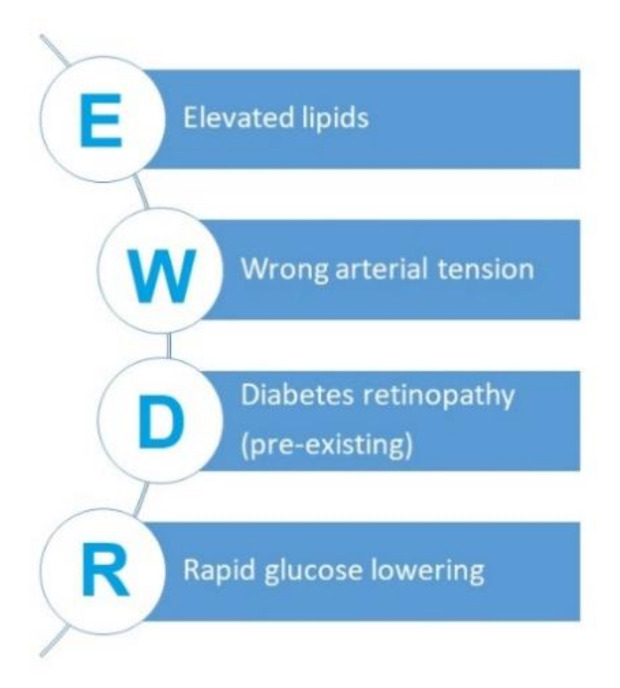
Modifiable risk factors for the development of EWDR- Early Worsening of Diabetic Retinopathy.

**Figure 2 jcm-10-00705-f002:**
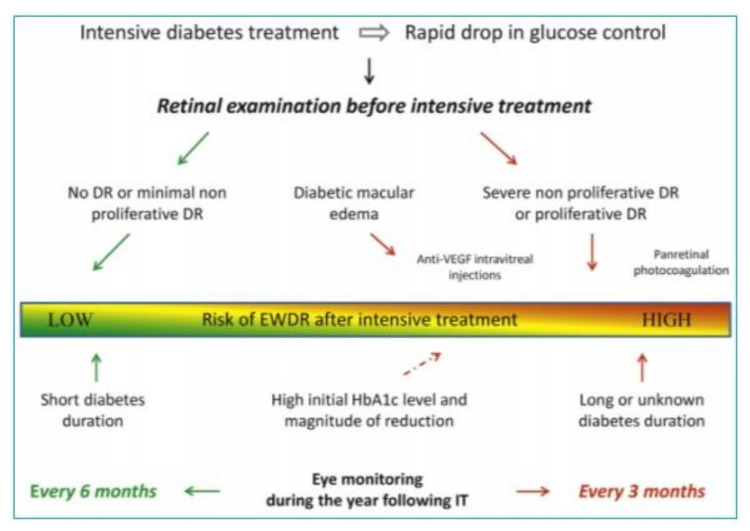
Eye monitoring and treatment algorithm associated with intensive diabetes treatment. DR: diabetic retinopathy; EWDR: early worsening diabetic retinopathy; VEGF: vascular endothelial growth factor; IT: intensive treatment [23].

**Figure 3 jcm-10-00705-f003:**
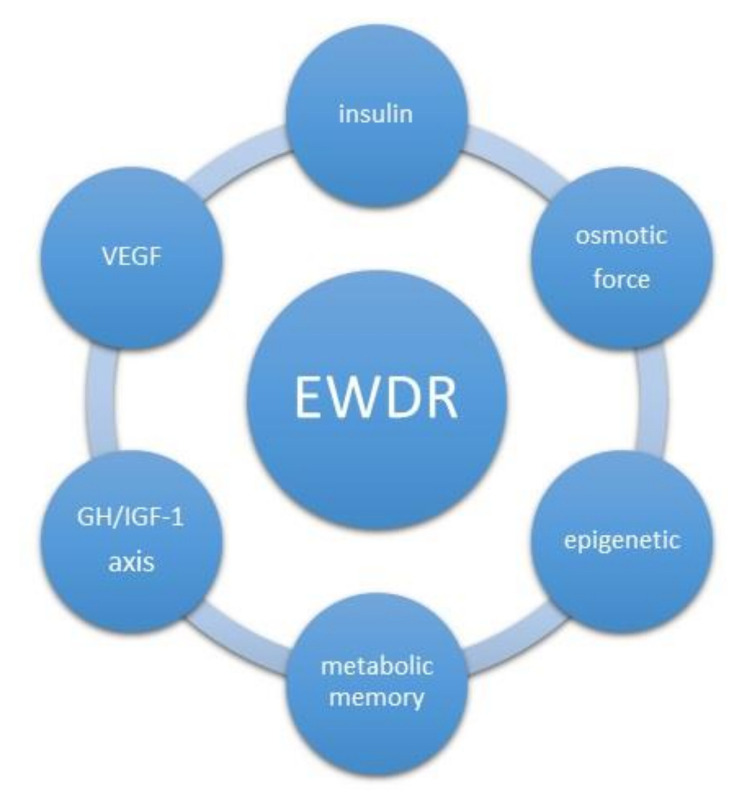
Potential mechanisms of Early Worsening of Diabetic Retinopathy development. GH/IGF1- growth hormone/insulin-like growth factor 1,VEGF- vascular endothelial growth factor

**Figure 4 jcm-10-00705-f004:**
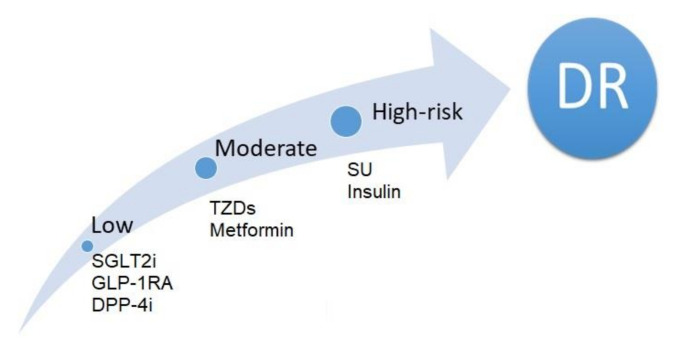
Comparisons of risk of diabetic retinopathy events associated with glucose lowering drugs in patients with DM. DR-diabetic retinopathy, SGLT2i- Sodium-dependent glucose transporters, GLP1- glucagon-like peptide 1,DPP4- dipeptidyl peptidase-4,TZD- thiazolidinediones, SU- Sulfonylurea.

## Data Availability

Publicly available datasets were analyzed in this study.

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
