# Peer review of "The Safety of Pharmacological and Surgical Treatment of Diabetes in Patients with Diabetic Retinopathy—A Review"

_jcm, 2021, doi:10.3390/jcm10040705_

Round 1
Reviewer 1 Report
As suggested authors made the necessary changes and this manuscript can be accepted in this form.
Author Response
Dear Sir or Madam,
Thank you very much for your review, the attention to our article and all the comments. We appreciate your help and the influence on the final version of the article.
Yours faithfully
Authors
Reviewer 2 Report
The revised manuscript is significantly improved as compared with the original version. Here are few remaining concerns I have:
- The authors list “elevated lipids” and “wrong arterial pressure” as risk factors for EWDR. However, from the literature the authors present, these two factors remain
- In 3.2, the authors discuss all the potential mechanisms of EWDR in one paragraph. This is quite difficult for the readers to follow. Please divide this into separate paragraphs.
Minor:
- Some of the phrasing is confusing and difficult to understand. For instance, the first sentence of section 3.1. The authors use “consist in” multiple times. It would be better to replace each of these instances with a phrase such as “characterized by” which is more commonly used.
- There are several typos. For instance: Page 2, line 59 - 60: “… in both metabolic controls. both the appearance …”
Author Response
Dear Sir or Madam,
Thank you for your re-review, we have provided answers to your comments below:
1. Subchapter 3.1. was intended to be a short introduction to EWDR. The information of the potential influence of blood pressure (marked in red) and lipids (marked in yellow) on the development of EWDR is short because the work focuses on the effect of antihyperglycaemic treatment on EWDR. We are considering writing another article about the effects of treating lipid disorders and hypertension on DR.
2. Thank you for noticing this, we have divided this subchapter into paragraphs, which will make it clear and easier to read.
Minor: 1, 2 –Corrected.
Thank you very much for your review, the attention to our article and all the comments. We appreciate your help and the influence on the final version of the article.
Yours faithfully,
Authors
Reviewer 3 Report
I appreciate the authors responding to all of my comments. All concerns, major and minor, have been addressed satisfactorily and sufficient visual representations have also been added, enabling authors to have a comprehensive understanding of the subject.
Author Response
Dear Sir or Madam,
Thank you very much for your review, the attention to our article and all the comments. We appreciate your help and the influence on the final version of the article.
Yours faithfully
Authors
This manuscript is a resubmission of an earlier submission. The following is a list of the peer review reports and author responses from that submission.
Round 1
Reviewer 1 Report
The manuscript entitled" The safety of pharmacological and surgical treatment of diabetes in patients with diabetic retinopathy-a review" -
- Although authors made an effort to elaborate the importance of pharmacological agents used for diabetes for DR patients- this manuscript doesn't emphasize anything new to the already existing database.
- In the literature database related to diabetes and DR there are many published articles which have elaborated in detail about the DR and treatment.
Reviewer 2 Report
Thank you for giving me an opportunity to reivew your manuscript.
It is well written and read well.
There are a couple of points I would like to suggest.
1. The safety of glucose lowering agents (including insulin) for early worsening of DR / DR depends on how efficacious they are, baseline HbA11c and the rate of reduction in HbA1c, presence of previous active DR. I think that should be discussed in detail.
2. With regard to GLP-1, Semaglutide needs a special attention because it is regarded as the most potent GLP-1.
3. It is very text heavy and a diagrame of new DR development theories would be good.
Reviewer 3 Report
This is overall a nice, well-written review of the literature on the impact of strategies in diabetes management on early worsening of diabetic retinopathy. The manuscript is up to date and comprehensive with a few important exceptions noted below. However, there are important improvements that can be made. Most notably, the authors do not introduce the concept of early worsening of diabetic retinopathy, even though majority of the studies the authors refer directly relate to this concept.
Major concerns:
- The focal point of this article is the well-known phenomenon of early worsening of diabetic retinopathy (EWDR), in which rapid improvement in blood glucose control results in paradoxical worsening of DR in the initial months. . However, the authors only briefly mention this critical concept and only tangentially discuss it in their section 3.2.1 on Insulin. There needs to be an entire section introducing and explaining this concept to provide a context for understanding the rest of the review.
- As part of this section, the authors should discuss about the risk factors of EWDR and the management and prevention of EWDR.
- The current review does not refer to an excellent review paper on this same topic, published in Diabetes & Metabolism in 2018 by S.Feldman-Billard. This previous article essentially covers most of what the current review addresses. It might be helpful for the author to emphasize what is new in their current review.
- The current review should cite an important study recently published in American Journal of Transplantation by Barbora Voglová, which investigates the early worsening of diabetic retinopathy after simultaneous pancreas and kidney transplantation.
- Section 3.1 (“New therapies on DR development”) needs revision and clarification. First, the authors should note that this section is focused on theories that relate to the early worsening phenomenon, not DR development as a whole, for which a multitude of theories exist. Second, these are not new theories, but have been postulated for quite some time, as indicated by some of the cited references. Third, the cited references are a bit scant, including abstracts, rather than full articles. For such an important section, the authors need to provide a stronger set of supportive references. Indeed, the authors indicate weaknesses of the osmotic theory. Without a greater deal of support, this section and its claims do not appear authoritative.
- This section on “New mechanisms on DR development” should be directly related to the early worsening phenomenon.” As it stands, the section does not seem to have a purpose in relation to the remainder of the review.
- There are several instances where English usage is not optimal and also typos. For example, page 2, line 83: “insulin has 83 been modified a number of times through genetic engineering”: this sentence is confusing and unclear.
Page 2, line 83: “1980 witnessed the first reports”
Page 82, line 89: “a 1892 observation concerning 18 patients treated with 89 personal insulin pumps.”
Reviewer 4 Report
The manuscript presents a comparison of various pharmacological and surgical treatments for diabetes and an evaluation of their safety and efficacy to treat Diabetic Retinopathy. This presents a promising summary for clinicians and researchers alike. The study is well structured, free-flowing, and easy to read. It reviews the literature comprehensively, though a few references need to be inserted. However, a section on the safety of epigenetic pharmacological treatment for DR is highly warranted though, given the vast number of recent clinical trials reflecting the high interest of clinicians and researchers in the same. Furthermore, the abstract mentions the treatments associated with the lowest and highest risks, but this is not mentioned anywhere else in the manuscript. A tabular summary reflecting these would be highly beneficial for the readers.
Major changes:
(1). The introduction needs to establish a stronger foundation for the DR pathology. A brief section describing the pathological signatures associated with the occurrence and progression of DR would suffice, or even a graphical or tabular representation. This would enable to reader to appreciate the results and discussion better.
(2). “Metabolic memory” is a critical component of DM, especially for DR as has been shown by multiple studies. Further, given the promise of “epi-drugs”, a section on the safety and efficacy of epigenetic drug targets in clinical trials for diabetes and their effect on DR is highly warranted and is surprisingly missing.
(3). The title of the manuscript leads the reader to believe it is discussing about the safety of both pharmacological and surgical treatments. However, the majority of the manuscript focusses on a wide variety of pharmacological treatments and the section on surgical treatments is minimal. It is understandable that surgical treatment being a relatively newer way of treating diabetes, will yield less literature but given the promise and widespread interest in the same, it would be better if the authors elaborate on the different types on metabolic surgeries and speculate on how their efficacy and safety can be improved with reference to DR.
(4) Tabular summary for the highest and lowest risk treatments associated with DR
Minor changes:
(Line#34): “At present, there are 463 million adult DM patients, which constitutes 9.3% of the world population.”. Please include reference.
(Line#40): “It can occur on every stage of DM, also when it is diagnosed, and it constitutes 80% of all cases of complete loss of sight in the DM population”. This sentence is not clear and would be better if more clarity is provided.
(Line # 57-61): “The osmotic force theory assumes that glucose is an osmotically active molecule. A change in the glucose concentration alters the osmotic pressure and water retention. Hyperglycaemia lowers the intra-vascular osmotic pressure, and the change in the osmotic gradient inside the cell compared to that inside the vessel results in fluid displacement, excessive stretching of the vessels and the formation of microaneurysms.” This section needs references.
(Line#226): “However, a post hock analysis” Please fix typing error.
(Line#328): Please fix typing error for “disbetes”.